# Management of Patients with Hypersensitivity to Platinum Salts and Taxane in Gynecological Cancers: A Cross-Sectional Study by the European Network of Young Gynaecologic Oncologists (ENYGO)

**DOI:** 10.3390/cancers16061155

**Published:** 2024-03-14

**Authors:** Tibor A. Zwimpfer, Esra Bilir, Khayal Gasimli, Andrej Cokan, Nicolò Bizzarri, Zoia Razumova, Joanna Kacperczyk-Bartnik, Tanja Nikolova, Andrei Pletnev, Ilker Kahramanoglu, Alexander Shushkevich, Aleksandra Strojna, Charalampos Theofanakis, Tereza Cicakova, Marcus Vetter, Céline Montavon, Gilberto Morgan, Viola Heinzelmann-Schwarz

**Affiliations:** 1Peter MacCallum Cancer Center, East Melbourne, VIC 3002, Australia; 2Gynecological Cancer Center, University Hospital Basel, 4031 Basel, Switzerland; celine.montavon@usb.ch (C.M.); viola.heinzelmann@usb.ch (V.H.-S.); 3Department of Biomedicine, University of Basel, 4031 Basel, Switzerland; 4Department of Global Health, Koç University Graduate School of Health Sciences, 34010 Istanbul, Turkey; 5Department of Obstetrics and Gynecology, University Hospitals Schleswig-Holstein, Campus Kiel, 24105 Kiel, Germany; 6Division of Gynecologic Oncology, Department of Gynecology and Obstetrics, Frankfurt Goethe University Hospital, Theodor-Stern-Kai 7, 60590 Frankfurt, Germany; dr.gasimli@yahoo.com; 7Department for Gynaecological and Breast Oncology, University Medical Centre Maribor, 2000 Maribor, Slovenia; cokan.andrej@gmail.com; 8UOC Ginecologia Oncologica, Dipartimento di Scienze della Salute della Donna, del Bambino e di Sanità Pubblica, Fondazione Policlinico Universitario Agostino Gemelli IRCCS, 00168 Rome, Italy; nicolo.bizzarri@yahoo.com; 9Department of Women’s and Children’s Health, Karolinska Institutet, 171 77 Stockholm, Sweden; zoia.razumova@ki.se; 10II Department of Obstetrics and Gynecology, Medical University of Warsaw, 02-091 Warsaw, Poland; asiakacperczyk@gmail.com; 11Department of Obstetrics and Gynecology, Klinikum Mittelbaden, Academic Teaching Hospital of Heidelberg University, 76532 Baden-Baden, Germany; nikolovatanja@gmail.com; 12Department of Obstetrics and Gynecology, University of Zielona Gora, 65-729 Zielona Gora, Poland; a.pletneff@gmail.com; 13Department of Gynecologic Oncology, Ankara City Hospital, 06680 Ankara, Turkey; ilkerkahramanoglu@gmail.com; 14Department of Obstetrics and Gynecology, Klinikum Friedrichshafen Medizin Campus Bodensee, 88048 Friedrichshafen, Germany; shushkevich.aliaksandr@medizincamus.de; 15Department of Gynecology and Gynecologic Oncology, Evangelische Kliniken Essen Mitte, 45136 Essen, Germany; ola.strojna@yahoo.com; 16Division of Gynaecologic Oncology, 1st Department of Obstetrics and Gynaecology, Alexandra Hospital, National and Kapodistrian University of Athens School of Health Sciences, 116 35 Athens, Greece; ch.theofanakis@gmail.com; 17ESGO Office, 110 00 Prague, Czech Republic; tereza.cicakova@esgo.org; 18Cancer Center, Cantonal Hospital Baselland, Medical University Clinic, 4410 Liestal, Switzerland; marcus.vetter@ksbl.ch; 19The OncoAlert Network, 22185 Lund, Sweden; gilmorgan@hotmail.com

**Keywords:** desensitization, platinum, taxane, hypersensitivity reaction, gynecologic cancer, chemotherapy

## Abstract

**Simple Summary:**

Hypersensitivity reactions (HSRs) to platinum and taxane are common, and desensitization can be used to complete the standard chemotherapy regimen with a good safety profile and high success rate. Our study showed that the use of desensitization for HSRs to taxane and platinum is low in clinical practice. Treatment of HSRs has been shown to be heterogeneous and dependent on the grade of the HSR. Guidelines for the treatment of HSRs to taxane and platinum in gynecologic cancers have been of great interest to clinicians. Our study highlights that the management of platinum and taxane HSRs in gynecological cancers could be standardized and that international guidelines need to be developed.

**Abstract:**

Platinum and taxane chemotherapy is associated with the risk of hypersensitivity reactions (HSRs), which may require switching to less effective treatments. Desensitization to platinum and taxane HSRs can be used to complete chemotherapy according to the standard regimen. Therefore, we aimed to investigate the current management of HSRs to platinum and/or taxane chemotherapy in patients with gynecologic cancers. We conducted an online cross-sectional survey among gynecological and medical oncologists consisting of 33 questions. A total of 144 respondents completed the survey, and 133 respondents were included in the final analysis. Most participants were gynecologic oncologists (43.6%) and medical oncologists (33.8%), and 77.4% (*n* = 103) were involved in chemotherapy treatment. More than 73% of participants experienced >5 HSRs to platinum and taxane per year. Premedication and a new attempt with platinum or taxane chemotherapy were used in 84.8% and 92.5% of Grade 1–2 HSRs to platinum and taxane, respectively. In contrast, desensitization was used in 49.4% and 41.8% of Grade 3–4 HSRs to platinum and taxane, respectively. Most participants strongly emphasized the need to standardize the management of platinum and taxane HSRs in gynecologic cancer. Our study showed that HSRs in gynecologic cancer are common, but management is variable and the use of desensitization is low. In addition, the need for guidance on the management of platinum- and taxane-induced HSRs in gynecologic cancer was highlighted.

## 1. Introduction

Platinum- and taxane-based chemotherapy is the standard of care for patients with advanced gynecologic cancers [1,2,3,4]. Platinum-based anticancer drugs, including carboplatin, cisplatin, and oxaliplatin, induce apoptosis in cancer cells through DNA damage by disrupting DNA repair mechanisms, whereas taxanes (paclitaxel, docetaxel, and cabazitaxel) induce apoptosis by suppressing microtubule dynamics, preventing proper spindle formation and blocking mitosis [5,6,7,8]. However, multiple treatments with the same drug, such as platinum and taxanes, may result in oncologic resistance and hypersensitivity reactions (HSRs). This has an impact on further treatment and outcomes by necessitating a switch to a chemotherapy regimen that is less effective and more toxic [9,10,11,12]. Premedication with antihistamines and corticosteroids is usually recommended for mild HSRs and is routinely administered for taxane- and platinum-based chemotherapy [10,11,12,13]. However, premedication is not effective in preventing more severe allergic reactions, particularly those to platinum salts [14,15].

Desensitization is the establishment of a temporary tolerance to a substance that previously triggered an HSR [16]. Desensitization protocols for HSRs to chemotherapeutic agents are based on a stepwise increase in infusion rates of highly diluted drug solutions, starting as slowly as a few micrograms per milliliter of a drug in the first hour, with increasing doses over several hours to a few days until the total cumulative therapeutic dose is achieved and tolerated [12,16,17,18,19,20]. Importantly, patients remain allergic to the drug and must be desensitized for each course of treatment, as desensitization induces tolerance to a drug only temporarily, depending on continued exposure [20]. It should be considered in patients with HSRs as a safe alternative to platinum salts and taxanes in the use of standard chemotherapy, aiming at the best therapeutic results according to international standards [17,20,21,22].

Platinum hypersensitivity affects approximately 5% of the general oncologic population and 8 to 16% of women with gynecologic cancers, and taxane hypersensitivity affects 10 to 13% of the general oncologic population [17,19,21,23,24]. This is of clinical importance and justifies an optimal strategy for the maintenance of treatment. At present, desensitization techniques are well established and there are international guidelines for their administration [18]. However, clinical effects of desensitization are rarely studied and few specialized centers offer desensitization as part of standard practice.

Therefore, the aims of our cross-sectional study were to (1) reveal the current management of HSRs to platinum and/or taxane chemotherapy in patients with gynecologic cancer and (2) determine if there is a need for standardizing the management of HSRs for best patient care.

## 2. Materials and Methods

### 2.1. Study Design

We conducted a cross-sectional survey among gynecological and medical oncologists. SurveyMonkey software (https://www.surveymonkey.com/home/) was used to create and distribute the questionnaire. The survey was posted online on the European Network of Young Gynaecological Oncologists (ENYGO), European Society of Gynaecological Oncology (ESGO), and Oncoalert social media channels, Furthermore, we sent emails to the ENYGO members and subscribers of the Oncoalert Newsletter. We collected data between April 2023 and September 2023.

### 2.2. Variables

The survey consisted of 33 questions in English, prepared and based on the expertise of gynecological and medical oncologists treating patients with gynecological cancers and validated by the ESGO Scientific Committee (Appendix A). The questionnaire contained four main sections including (1) general demographic information (nine questions), (2) HSRs and platinum-based chemotherapy (11 questions), (3) HSRs and chemotherapy with taxane (11 questions), and (4) future direction of HSRs and desensitization in gynecological cancer (two questions). HSR Grade was defined as Common Terminology for Adverse Events (CTCAEs) Grades 1–4. Briefly, CTCAE 1, 2, 3, and 4 refer to mild, moderate, severe, and life-threatening adverse events, respectively.

### 2.3. Data Sources/Measurement

The software SurveyMonkey was used for questionnaire creation and distribution. Data were collated from members and followers of European Network of ENYGO and ESGO (newsletter including HSR Survey received = 1938 and opened = 106). Additionally, the survey was placed online on ENYGO and Oncoalert social media channels (opened = 53).

### 2.4. Statistical Analyses

Descriptive statistics are presented as counts and frequencies for categorical data and medians (range) for metric or ordinal variables. Cases with median *p*-values correspond to the Kruskall–Wallis tests, and cases with categorical data *p*-values correspond to Fisher’s exact tests. *p*-values of group comparisons correspond to log-rank tests. A *p* value < 0.05 was considered significant. All analyses were performed using the Statistical Package for the Social Sciences (SPSS) for Macintosh, Version 28.0 (IBM Corp., Armonk, NY, USA).

## 3. Results

### 3.1. Demographic Data

A total of 144 respondents from 33 countries completed the survey. The final analysis included 133 respondents, of whom 79 administered platinum-based and 67 taxane-based chemotherapy. We excluded respondents who were only involved in the treatment of breast cancer (*n* = 6), respondents who did not treat gynecological cancers (*n* = 5), and one respondent who was a nurse. The countries with the highest participation were Switzerland (10.5%), Italy (9%), and Germany (8.3%) (Table 1, Appendix A). The genders of the participants were balanced, with 54.9% female and 45.1% male and a mean age of 38 years. The majority of participants were gynecological oncologists (43.6%) and medical oncologists (33.8%). Seventy-six participants (57.1%) worked mainly in university hospitals and 103 participants (77.4%) were involved in chemotherapy treatment. The clinical experience of the participants was well balanced, with 32.3% having less than 5 years, 33.8% 5–10 years, and 33.8% more than 10 years. Detailed demographic characteristics of these participants are shown in Table 1.

### 3.2. HSR and Platinum-Based Chemotherapy

Out of the 79 participants who administered platinum-based chemotherapy, more than half of them treated more than 100 gynecological cancer patients per year with platinum-based chemotherapy (Table 2). The majority (73.4%) of participants experienced more than five platinum HSRs per year. In 84.8% of Grade 1 and 2 HSRs (according to CTCAE), participants used premedication with antihistamines/steroids and made new attempts with standard infusions of platinum-based chemotherapy. However, 41.8% used desensitization in these patients, only 15.2% stopped chemotherapy, and 8.9% changed the treatment regimen. In contrast, in cases with Grade 3–4 HSRs, 35.4% of the participants suspended chemotherapy, while 34.1% changed the regimen. There was a minimal increase in the use of desensitization in patients with Grades 3–4 HSRs compared to patients with Grade 1–2 HSRs (49.4% vs. 41.8%). Desensitization was mainly performed by medical oncologists (40.5%) and allergologists (27.8%) in their own clinics. Sixty-seven percent of the participants were able to continue platinum-based chemotherapy after tolerance was achieved in more than 50% of the cases. However, 45.6% of participants experienced one or more critical events during tolerance induction, mainly due to recurrent HSR CTCAE Grade 1–2 (47.2%) and Grade 3–4 (66.7%).

### 3.3. HSR and Taxane-Based Chemotherapy

A total of 67 participants administered taxane-based chemotherapy, with the majority treating more than 100 gynecological cancer patients per year and regularly experiencing HSRs in their patients (Table 2). Premedication with antihistamines and steroids together with a retry of the standard infusion was the main mode of action for Grade 1–2 HSRs (92.5%). In contrast, in Grade 3–4 HSRs, the mode of action was balanced (Table 2). When desensitization was used, the majority were able to continue with standard taxane treatment. Desensitization was performed by medical oncologists (34.3%), allergologists (23.9%), and gynecological oncologists (10.4%). Of the participants, 52% never experienced a critical incident during tolerance induction, but 25.4% experienced more than one. The main reasons for critical incidents are HSRs to taxane CTCAE Grades 1 and 2 (34.5%) and Grade 3–4 (62.1%).

### 3.4. Desensitization of HSRs in Gynecological Cancers

The majority (53.3%) of participants without experience in chemotherapy treatment were not aware of desensitization of HSRs to platinum- and taxane-based chemotherapy, whereas the majority of participants involved in chemotherapy treatment were aware of desensitization. However, participants strongly emphasized the need to standardize the management of platinum and taxane HSRs in gynecological cancer and to develop international guidelines, regardless of their involvement in chemotherapy treatment (Table 3).

### 3.5. Management of Hypersensitivity Reactions Based on Length of Clinical Practice Experience

No significant difference was evident in the results of the questions comparing participants with less than five years, five to ten years, or more than ten years of clinical practice experience (Figure 1). However, participants with more than ten years of experience were more likely to report experiencing <5 or 5–10 HSRs to taxane per year than the participants with five to ten years of experience or less than 5 years of experience (*p* = 0.38, 8 vs. 6 vs. 4 and 5 vs. 4 vs. 1, respectively) (Figure 1). In addition, there was a trend that the longer the experience in clinical practice, the more often the participants did not think there was a need for standardization and guidelines for managing HSRs to taxane and platinum (*p* = 0.26) (Figure 2).

## 4. Discussion

The majority of the participants experienced a high frequency of HSRs to taxane and platinum in their clinical practice, with more than five HSRs per year. Management of HSRs is heterogeneous and depends on the grade of the HSR. Overall, we found that the use of desensitization for HSRs to taxane and platinum in clinical practice is low at less than 50% and guidelines for the treatment of HSRs to taxane and platinum in gynecological cancers were of great interest to clinicians, regardless of their experience with chemotherapy.

Clinicians treating gynecological cancers regularly experience HSRs to platinum- and taxane-based chemotherapy (8–16% and 10–13%, respectively) [11,12,17,19,21,23,24], which is also due to the fact that more lines of treatment are being used in gynecological cancer than 1–2 decades ago [25]. Additionally, real HSR rates are likely to be underestimated, as oncologists often report only severe reactions [10,26]. Our study showed a high frequency of HSRs, with more than 73% of the participants treating more than five patients with HSRs to platinum- and taxane-based chemotherapy, which emphasizes the need to find a strategy to maintain the optimal treatment regimen in this large group of patients.

The treatment strategy for HSRs to taxanes and platinum often depends on its Grade. For mild HSRs, premedication with antihistamines and corticosteroids is typically recommended and routinely used [10,11,12,13]. This is well represented in our study, with more than 84.8% and 92% of patients with Grade 1–2 HSRs receiving platinum and taxane, respectively, regularly premedicated with antihistamines and corticosteroids. However, premedication is ineffective in preventing more severe HSRs (Grade 3–4) to platinum and taxane, and therefore, desensitization should be considered to continue standard chemotherapy for the best therapeutic outcome in these patients [14,15,17,20,21,22].

The results of our survey showed a higher use of desensitization in Grade 3–4 HSRs compared to Grade 1–2 HSRs with platinum and taxane (49.4% and 48.8% vs. 41.8% and 20.9%, respectively). Moreover, 31.6% and 44.8% of the participants regularly premedicate their patients with antihistamines and corticosteroids for Grade 3–4 HSRs to platinum and taxane, respectively. In addition, there is a high rate of switching to another treatment regimen, with 34.1% for platinum-based and 44.8% for taxane-based chemotherapy. This may be explained by the finding that only 67.1% of patients after platinum desensitization and 58.2% of patients after taxane desensitization had a high likelihood (>50%) of continuing platinum- or taxane-based chemotherapy. In addition, there was a high incidence of critical events (40%) during the desensitization process. This is in contrast to what is known about the safety of desensitization procedures and their management [17,18,19,20,21,22]. This is important to address, as an improved outcome for overall survival has been demonstrated in hypersensitive patients receiving carboplatin desensitization compared to non-hypersensitive patients in relapsed ovarian cancer, independent of the germline *BRCA* status [27]. However, standard desensitization protocols are not widely accepted or used, in part because they can be time consuming and in part because there are several different protocols available [17,20,21,22]. One way to address this important issue is to standardize the management of platinum and taxane HSRs in gynecological cancer by developing international guidelines. This was particularly emphasized by participants with (59.2%) and without (83.3%) experience in the medical treatment of gynecological cancer patients.

This survey provides a global representation of participants and their current management of platinum and taxane HSRs in gynecological cancer. An advantage is the direct feedback from clinicians regularly confronted with HSRs in their daily clinical practice on their awareness and views on this topic. However, the small cohort size is a weakness of this study and limits the statistical power of the results. This is an anticipated problem with survey studies, especially when the target group are clinicians with a heavy workload and limited time to complete a survey. However, a variety of methods were used to distribute the survey, including the official social media channels of ENYGO, ESGO, and Oncoalert, and distribution of the survey via an email system to the ENYGO, ESGO, and Oncoalert databases. However, despite the variety of methods used, there was still a low response rate of 5%. Possible solutions to this problem could be to distribute the survey directly at congresses and workshops or to distribute it on a personal level, which could improve the response rate. Additionally, the fact that 23% of the respondents were not involved in chemotherapy treatment is a major limitation, as this could bias the results. To account for this, we analyzed the results of the questionnaire on HSRs to platinum- and taxane-based chemotherapy only for those respondents involved in chemotherapy treatment.

Currently, only a limited number of cancer centers have established desensitization as part of their standard practice. However, desensitization protocols for patients with taxane and platinum HSRs are available and recommended [10,11,12,18,19]. Knowledge of desensitization procedures in gynecological oncology could be optimized by regular analysis and management of successful tolerance induction to platinum and taxanes in patients with HSRs. This is important to achieve optimal treatment in accordance with international standards [14,15]. However, since the goal is to provide the best treatment within the recommended timeframe, it is also important not to delay planned chemotherapy for desensitization. For this reason, patients who develop HSRs should be seen and tested within one to two weeks of the reaction. This underscores the importance of a multidisciplinary approach to gynecologic cancer care in specialized centers and adherence to clinical guidelines, which ensures a better prognosis and quality of life for patients [28,29,30,31,32,33,34,35]. In particular, it has been shown that the collaboration of different experts leads to an increased awareness of potential treatments and a better evaluation of diagnostic–therapeutic areas beyond their own competence, which ultimately improves the effectiveness of treatments [28,29,30,31,36]. Our study shows the willingness of the participants to use the guidelines for the treatment of HSRs when they become available.

## 5. Conclusions

Our cross-sectional survey showed that HSRs in gynecological cancer are common, but management is variable with low use of desensitization. In addition, clinical practitioners emphasized the need for standardization and guidelines for the management of HSRs to platinum and taxane in gynecological cancer.

## Figures and Tables

**Figure 1 cancers-16-01155-f001:**
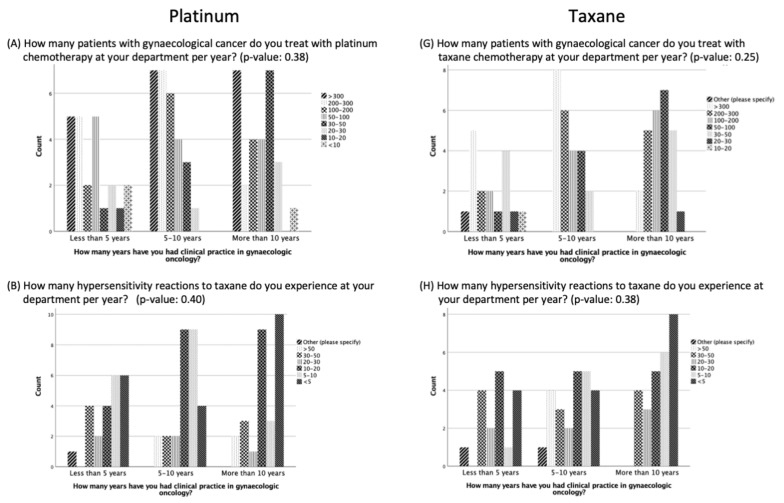
Management of hypersensitivity reactions to platinum and taxane based on duration of clinical practice experience.

**Figure 2 cancers-16-01155-f002:**
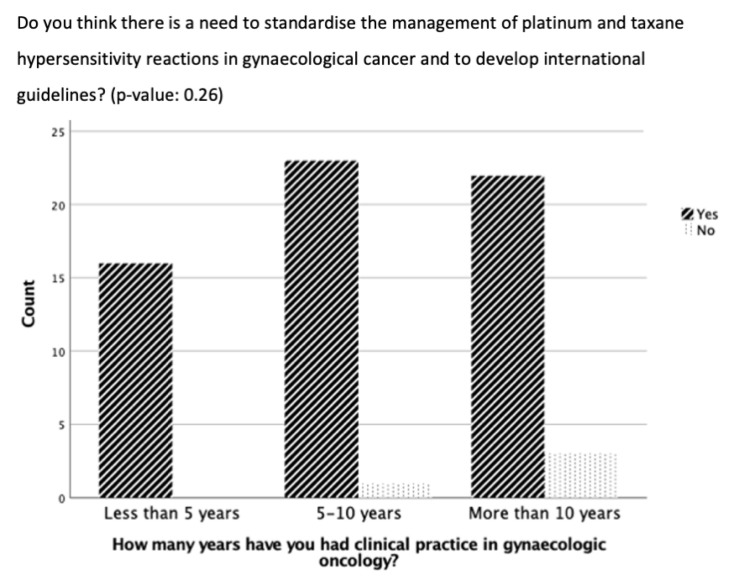
Need for standardization of desensitization for hypersensitivity reactions in gynecological cancer based on duration of clinical practice experience.

**Table 1 cancers-16-01155-t001:** Demographic representation of the participants.

Variable	Number (*n* = 133, %)
Gender	
Female	73 (54.9)
Male	60 (45.1)
Age (years) (median, IQR)	38 (35, 43)
Country	
Switzerland	14 (10.5)
Italy	12 (9.0)
Germany	11 (8.3)
India	9 (6.8)
Azerbaijan	8 (6.0)
Slovenia	8 (6.0)
Setting *	
University Hospital	76 (57.1)
Regional Hospital	16 (12.0)
Public Hospital	24 (18.0)
Private Hospital	22 (16.5)
Private practice	4 (3.0)
Other	3 (2.3)
Specialty	
Gynecologic oncologist	58 (43.6)
Gynecologist	18 (13.5)
Medical oncologist	45 (33.8)
Radiation oncologist	8 (6.0)
Other (please specify)	4 (3.0)
Types of gynecological cancer treated *	
Ovarian	123 (92.5)
Cervical	126 (94.7)
Vulvar	104 (78.2)
Vaginal	100 (75.2)
Corpus/Endometrium	114 (85.7)
Breast	62 (46.6)
Other	5 (3.8)
Clinical practice in gynecologic oncology	
Less than 5 years	43 (32.3)
5–10 years	45 (33.8)
More than 10 years	45 (33.8)
Involved in chemotherapy treatment	
Yes	103 (77.4)
No	30 (22.6)

* Multiple answers can be selected. *n* = Number, IQR = interquartile range.

**Table 2 cancers-16-01155-t002:** Results of the questions about hypersensitivity reactions and platinum- and taxane-based chemotherapy.

Questions	Platinum *n* = 79 (*n*, %)	Taxane *n* = 67 (*n*, %)
Gynecological cancers treated with platinum/taxane per year		
>300	19 (24.1)	15 (22.4)
200–300	14 (17.7)	13 (19.4)
100–200	12 (15.2)	12 (17.9)
50–100	13 (16.5)	12 (17.9)
30–50	11 (13.9)	11 (16.4)
20–30	6 (7.6)	2 (3.0)
10–20	1 (1.3)	1 (1.5)
<10	3 (3.8)	1 (1.5)
HSRs to platinum/taxane per year		
>50	4 (5.1)	4 (6.0)
30–50	9 (11.4)	11 (16.4)
20–30	5 (6.3)	7 (10.4)
10–20	22 (27.8)	15 (22.4)
5–10	18 (22.8)	12 (17.9)
<5	20 (25.3)	16 (23.9)
Other	1 (1.3)	2 (3.0)
HSRs to platinum/taxane CTCAE Grade 1–2 *		
Premedication with antihistamines/steroids and new attempt with standard infusion	67 (84.8)	62 (92.5)
Suspension of the chemotherapy	12 (15.2)	11 (16.4)
Change the chemotherapy to, e.g., Oxaliplatin	7 (8.9)	11 (16.4)
Tolerance induction (stepwise increase of infusion rate of highly diluted platinum dilution)	33 (41.8)	14 (20.9)
Other	3 (3.8)	2 (2.9)
HSR to platinum/taxane CTCAE Grade 3–4 *		
Premedication with antihistamines/steroids and new attempt with standard infusion	25 (31.6)	30 (44.8)
Suspension of the chemotherapy	28 (35.4)	25 (37.3)
Change the chemotherapy to, e.g., Oxaliplatin	27 (34.1)	30 (44.8)
Tolerance induction (stepwise increase of infusion rate of highly diluted platinum dilution)	39 (49.4)	28 (41.8)
Other	0 (0.0)	3 (4.5)
Performing tolerance induction of platinum/taxane		
Yes ^†^, at our clinic	46 (58.2)	37 (55.2)
No, but I referred the patient to another clinic	13 (16.5)	8 (11.9)
No	20 (25.3)	21 (31.3)
Other	0 (0.0)	1 (1.5)
^†^ If yes, who performs the tolerance induction of platinum/taxane *		
Allergologist	22 (47.8)	16 (43.2)
Medical oncologist	32 (69.56)	23 (62.12)
Specialist for internal medicine	2 (4.3)	0 (0.0)
Gynecologic oncologist	12 (26.1)	7 (18.9)
Other	5 (10.8)	4 (10.8)
How many times can you continue the chemotherapy after tolerance induction of platinum/taxane		
every time	15 (19.0)	11 (16.4)
>50%	38 (48.1)	28 (41.8)
<50%	18 (22.8)	9 (13.4)
never	8 (10.1)	13 (19.4)
Other	0 (0.0)	6 (9.0)
Experience of a critical incident event in the course of the tolerance induction of platinum/taxane		
Yes ^§^, more than once	20 (25.3)	17 (25.4)
Yes ^§^, once	16 (20.3)	12 (17.9)
No	43 (54.4)	35 (52.2)
Other (please specify)	0 (0.0)	3 (4.5)
^§^ If yes, the reason(s) *		
HSR to platinum/taxane CTCAE Grade 1–2	17 (47.2)	10 (34.5)
HSR to platinum/taxane CTCAE Grade 3–4	24 (66.7)	18 (62.1)
Death	2 (5.6)	1 (3.4)
Patient not informed about the risks of tolerance induction	2 (5.6)	1 (3.4)
Other	4 (11.1)	2 (6.9)

* Multiple answers can be selected. ^†^ Link the answer to the follow-up question. ^§^ Link the answers to the follow-up question. *n* = Number, HSR = Hypersensitivity reaction, CTCAE = Common Terminology for Adverse Events.

**Table 3 cancers-16-01155-t003:** Results of the question on awareness and need for standardization of desensitization for hypersensitivity reactions in gynecological cancer.

Question	Not Involved in Chemotherapy Treatment *n* = 30 (%)	Involved in Chemotherapy Treatment *n* = 103 (%)
Aware of the possibility of desensitization of patients with HSRs to platinum/taxane prior to this survey		
Yes	12 (40.0)	57 (55.3)
No	16 (53.3)	7 (6.8)
Other	0 (0)	2 (1.9)
No answer	2 (2.7)	37 (35.9)
A need to standardize the management of platinum and taxane HSRs in gynecological cancer and to develop international guidelines?		
Yes	25 (83.3)	61 (59.2)
No	2 (6.7)	4 (3.9)
No answer	3 (10.0)	38 (36.9)

HSR = Hypersensitivity reaction.

## Data Availability

The datasets that have been used and/or analyzed during the current study are available from the corresponding author on reasonable request.

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
