# Peer review of "Management of Patients with Hypersensitivity to Platinum Salts and Taxane in Gynecological Cancers: A Cross-Sectional Study by the European Network of Young Gynaecologic Oncologists (ENYGO)"

_cancers, 2024, doi:10.3390/cancers16061155_

Round 1
Reviewer 1 Report
Comments and Suggestions for Authors
1. Subgroup analysis may be needed between the doctors with patient who face allergy effect or not, is the treatment concept be changed?
2. It will be better to simply introduce the desensitization protocols most used for patients with taxane and platinum HSRs in the common practice.
3. Is the critical condition of desensitization easily managed? Is there any legal problem raised during the procedure? How did you discuss the inform consent for the desensitization protocols?
4. What is the major factor to decide whether to change the chemotherapy regimen or perform the desensitization protocol?
Author Response
Thank you for the opportunity to revise our manuscript “Management of patients with hypersensitivity to platinum salts and taxane in gynecological cancers: a cross-sectional study by the European Network of Young Gynaecologic Oncologists (ENYGO)". We appreciate your careful review and constructive suggestions. We believe that the manuscript is substantially improved after the suggested changes have been made.
Following you can find our responses in italics, including how and where the text was modified. Changes in the manuscript are highlighted in blue. The revision has been developed in consultation with all co-authors, and each author has agreed to the final form of this revision. The consent form signed by each author remains valid.
Comment 1: Subgroup analysis may be needed between the doctors with patient who face allergy effect or not, is the treatment concept be changed?
Response 1: Thank you for this great suggestion. We would be very interested in this subgroup analysis, but almost all physicians have patients with HSR and therefore unfortunately we could not perform this subgroup analysis within our study.
Comment 2: It will be better to simply introduce the desensitization protocols most used for patients with taxane and platinum HSRs in the common practice.
Response 2: Thank you for pointing this out. We have now introduced the principles of desensitization therapy and desensitization protocols in the Introduction (page 2, lines 85-91).
“Desensitization is the establishment of a temporary tolerance to a substance that previously triggered HSR (16). Desensitization protocols for HSRs to chemotherapeutic agents are based on a stepwise increase in infusion rates of highly diluted drug solutions, starting as slowly as a few micrograms per milliliter of drug in the first hour, with increasing doses over several hours to a few days until the total cumulative therapeutic dose is achieved and tolerated (12, 16-20). Importantly, patients remain allergic to the drug and must be desensitized for each course of treatment, as desensitization induces tolerance to a drug only temporarily, depending on continued exposure (20). It should be considered in patients with HSR as a safe alternative to platinum salts and taxanes in the use of standard chemotherapy, aiming at the best therapeutic results according to international standards (17, 20-22).”
Comment 3: Is the critical condition of desensitization easily managed? Is there any legal problem raised during the procedure? How did you discuss the inform consent for the desensitization protocols?
Response: 3: Thank you for pointing this out. We have discussed this important issue in the discussion (Page 12, lines 265-268) and the introduction (page 2, lines 85-91). The desensitization procedures are safe and well tolerated and an informed consent is used as with regular chemotherapy treatments. However, standard desensitization protocols are not widely accepted or used, in part because they can be time consuming and in part because there are several different protocols available.
“In addition, there was a high incidence of critical events (40%) during the desensitization process. This is in contrast to what is known about the safety of desensitization procedures and their management (17-22). This is important to address as an improved outcome for overall survival has been demonstrated in hypersensitive patients receiving carboplatin desensitization compared to non-hypersensitive patients in relapsed ovarian cancer, independent of germline BRCA status (27). However, standard desensitization protocols are not widely accepted or used, in part because they can be time consuming and in part because there are several different protocols available (17, 20-22). One way to address this important issue is to standardise the management of platinum and taxane HSR in gynecological cancer by developing international guidelines. This was particularly emphasized by participants with (59.2%) and without (83.3%) experience of medical treatment of gynecological cancer patients.”
Comment 4: What is the major factor to decide whether to change the chemotherapy regimen or perform the desensitization protocol?
Response 4: Thank you for this comment. We believe it is for the same reason as stated above. “However, standard desensitization protocols are not widely accepted or used, in part because they can be time consuming and in part because there are several different protocols available (17, 20-22). (page 12, lines 265-268)
Reviewer 2 Report
Comments and Suggestions for Authors
The manuscript submitted for review concerns the interesting and important topic of desensitization of patients with HSR during ovarian cancer treatment with platinum and taxanes. This is currently a significant problem in light of postponing or discontinuing treatment with the most effective cytostatics or changing chemotherapeutics.
The manuscript describing the conducted research has many advantages: it is multicenter, international, up-to-date, created by young people, correctly planned, and neatly written. For me, who deals with the treatment of malignant tumors in gynecology, it was exceptionally interesting and engaging.
I have no critical comments. The manuscript is suitable for publication in its current form. I enjoyed reading it and learned a lot.
Author Response
We would like to thank you very much for this recognition and your thorough review.Reviewer 3 Report
Comments and Suggestions for Authors
Zwipfer and colleagues presented a research article discussing the knowledge on hypersensitivity reactions (HSRs) to platinum and taxane and on desensitization therapy. For this purpose, the authors proposed a questionnaire, however, the study indicates low usage of desensitization in clinical practice for taxane and platinum HSRs highlighting the need for other guidelines. Overall, the manuscript is interesting, however, there are some issues that the authors have to address before publication:
1) In the Introduction section, before describing desensitization treatment, you should briefly describe the mechanisms of action of both drugs. For this purpose, please see:
- https://doi.org/10.1016/j.cell.2023.02.038
- https://doi.org/10.3390/cancers12113323
- https://doi.org/10.1016/j.ejphar.2014.07.025
2) Is the proposed questionnaire validated by the international societies on gynecological cancer? Please clarify;
3) The rate between received and opened surveys is very low (about 5%). How do you explain this data? In the Discussion section, you have to propose solutions for this issue;
4) Please clarify the following issue:
Opened survey: 106
Total respondentsts: 144.
There is an incongruence in these data;
5) In the Discussion or Introduction section, you have to better describe the principles of desensitization therapy;
6) The present study highlighted a key limiting aspect in the management of gynecological tumors that is the lack of standardized guidelines for the treatment of HSR. In the Discussion section, you have to add a brief paragraph describing the importance of a multidisciplinary approach in the treatment of gynecological tumors to better manage these pathologies as well as the importance of consensus conferences to establish new treatment guidelines. For this purpose, please see:
- https://doi.org/10.3892/ijo.2021.5233
- https://doi.org/10.2147/CMAR.S220976
- https://doi.org/10.1016/S1470-2045(22)00139-5
- https://doi.org/10.1093/annonc/mdz062
Author Response
Thank you for the opportunity to revise our manuscript “Management of patients with hypersensitivity to platinum salts and taxane in gynecological cancers: a cross-sectional study by the European Network of Young Gynaecologic Oncologists (ENYGO)". We appreciate your careful review and constructive suggestions. We believe that the manuscript is substantially improved after the suggested changes have been made.
Following you can find our responses in italics, including how and where the text was modified. Changes in the manuscript are highlighted in blue. The revision has been developed in consultation with all co-authors, and each author has agreed to the final form of this revision. The consent form signed by each author remains valid.
Comment 1: In the Introduction section, before describing desensitization treatment, you should briefly describe the mechanisms of action of both drugs. For this purpose, please see:
- https://doi.org/10.1016/j.cell.2023.02.038
- https://doi.org/10.3390/cancers12113323
- https://doi.org/10.1016/j.ejphar.2014.07.025
Response 1: We are very grateful for this comment. We have adapted the introduction accordingly and briefly described the mechanism of action for platinum and taxanes (page 2, lines 72-76).
“Platinum- and taxane-based chemotherapy is the standard of care for patients with advanced gynecologic cancers (1-4). Platinum-based anticancer drugs, including carboplatin, cisplatin, and oxaliplatin, induce apoptosis in cancer cells through DNA damage by disrupting DNA repair mechanisms, whereas taxanes (paclitaxel, docetaxel, and cabazitaxel) induce apoptosis by suppressing microtubule dynamics, preventing proper spindle formation and blocking mitosis (5-8). However, multiple treatments with the same drug, such as platinum and taxanes, may result in oncologic resistance and hypersensitivity reactions (HSR). This has an impact on further treatment and outcome by necessitating a switch to a chemotherapy regimen that is less effective and more toxic (9-12). Pre-medication with antihistamines and corticosteroids is usually recommended for mild HSR and routinely administered for taxane and platinum-based chemotherapy (10-13). However, premedication is not effective in preventing more severe allergic reactions, particularly those to platinum salts (14, 15).”
Comment 2: Is the proposed questionnaire validated by the international societies on gynecological cancer? Please clarify;
Response 2: Thank you for pointing that out. In fact, the survey has been validated by the members of the ESGO Scientific Committee, co-chaired by Professor Mansoor Mirza and Jonathan Ledermann. We have included this information in the Methods section "Variables" (page 3, lines 110-111).
“Variables
The survey consisted of 33 questions in English, prepared and based on the expertise of gynecological and medical oncologists treating patients with gynecological cancers and validated by the ESGO Scientific Committee (Supplementary file 1). The questionnaire contained four main sections including 1) general demographic information (nine questions), 2) HSR and platinum-based chemotherapy (11 questions), 3) HSR and chemotherapy with taxane (11 questions), and 4) future direction of HSR and desensitization in gynecological cancer (two questions). HSR Grade was defined as Common Terminology for Adverse Events (CTCAE) Grade 1-4. Briefly, CTCAE 1, 2, 3, and 4 refer to mild, moderate, severe, and life-threatening adverse events, respectively.”
Comment 3: The rate between received and opened surveys is very low (about 5%). How do you explain this data? In the Discussion section, you have to propose solutions for this issue;
Response 3: Thank you for pointing this out. This is an important and well-known issue in surveys, especially when the target audience is clinicians with a heavy workload and limited time to complete a survey (patient surveys often have a better response rate). We have adjusted the discussion accordingly with possible solutions for future surveys (page 12, lines 281-284).
“This survey provides a global representation of participants and their current management of platinum and taxane HSR in gynecological cancer. An advantage is the direct feedback from clinicians regularly confronted with HSR in their daily clinical practice on their awareness and views on this topic. However, the small cohort size is a weakness of this study and limits the statistical power of the results. This is an anticipated problem with survey studies, especially when the target group are clinicians with a heavy workload and limited time to complete a survey. However, a variety of methods were used to distribute the survey, including the official social media channels of ENYGO, ESGO, and Oncoalert, and distribution of the survey via an email system to ENYGO, ESGO, and Oncoalert databases. However, despite the variety of methods used, there was still a low response rate of 5%. Possible solutions to this problem could be to distribute the survey directly at congresses and workshops or to distribute it on a personal level, which could improve the response rate. Additionally, the fact that 23% of the respondents were not involved in chemotherapy treatment is a major limitation, as this could bias the results. To account for this, we analysed the results of the questionnaire on HSR to platinum and taxane-based chemotherapy only for those respondents involved in chemotherapy treatment.”
Comment 4: Please clarify the following issue:
Opened survey: 106
Total respondentsts: 144.
There is an incongruence in these data;
Response 4: Thank you for pointing this out. We apologize for the confusion. 106 opened the survey from the newsletter sent to members and followers of ENYGO and ESGO. In addition, 53 opened the survey after it was posted online on ENYGO and Oncoalert social media channels. We have adjusted this information accordingly in the Methods section "Data Sources/Measurement" (page 3, line 130).
"Data sources/measurement
The software SurveyMonkey was used for questionnaire creation and distribution. Data were collated from members and followers of European Network of ENYGO and ESGO (Newsletter including HSR Survey received=1938 and opened=106). Additionally, the survey was put online on ENYGO and Oncoalert social media channels (opened=53)."
Comment 5: In the Discussion or Introduction section, you have to better describe the principles of desensitization therapy;
Response 5: Thank you for this comment. We have now introduced the principles of desensitization therapy in the Introduction (page 2, lines 85-91).
“Desensitization is the establishment of a temporary tolerance to a substance that previously triggered HSR (16). Desensitization protocols for HSRs to chemotherapeutic agents are based on a stepwise increase in infusion rates of highly diluted drug solutions, starting as slowly as a few micrograms per milliliter of drug in the first hour, with increasing doses over several hours to a few days until the total cumulative therapeutic dose is achieved and tolerated (12, 16-20). Importantly, patients remain allergic to the drug and must be desensitized for each course of treatment, as desensitization induces tolerance to a drug only temporarily, depending on continued exposure (20). It should be considered in patients with HSR as a safe alternative to platinum salts and taxanes in the use of standard chemotherapy, aiming at the best therapeutic results according to international standards (17, 20-22).”
Comment 6: The present study highlighted a key limiting aspect in the management of gynecological tumors that is the lack of standardized guidelines for the treatment of HSR. In the Discussion section, you have to add a brief paragraph describing the importance of a multidisciplinary approach in the treatment of gynecological tumors to better manage these pathologies as well as the importance of consensus conferences to establish new treatment guidelines. For this purpose, please see:
- https://doi.org/10.3892/ijo.2021.5233
- https://doi.org/10.2147/CMAR.S220976
- https://doi.org/10.1016/S1470-2045(22)00139-5
- https://doi.org/10.1093/annonc/mdz062
Response 6: Thank you for this suggestion. We have adapted the discussion accordingly and added a paragraph regarding the multidisciplinary approach in the treatment of gynaecological cancers (page 12 and 13, lines 298-304).
“Currently, only a limited number of cancer centres have established desensitization as part of their standard practice. However, desensitization protocols for patients with taxane and platinum HSRs are available and recommended (10-12, 18, 19). Knowledge of desensitization procedures in gynecological oncology could be optimized by regular analysis and management of successful tolerance induction to platinum and taxanes in patients with HSR. This is important to achieve an optimal treatment in accordance with international standards (14, 15). However, since the goal is to provide the best treatment within the recommended timeframe, it is also important not to delay planned chemotherapy for desensitization. For this reason, patients who develop an HSR should be seen and tested within one to two weeks of the reaction. This underscores the importance of a multidisciplinary approach to gynecologic cancer care in specialized centers and adherence to clinical guidelines, which ensures better prognosis and quality of life for patients (28-35). In particular, it has been shown that the collaboration of different experts leads to an increased awareness of potential treatments and a better evaluation of diagnostic-therapeutic areas beyond their own competence, which ultimately improves the effectiveness of treatments (28-31, 36). Our study shows the willingness of the participants to use the guidelines for the treatment of HSR when they become available.”
Round 2
Reviewer 3 Report
Comments and Suggestions for Authors
The authors well addressed all my previous comments. The revised version of the manuscript appears now more detailed and complete and can be approved for publication.